# Robust Adaptive Heading Control for a Ray-Type Hybrid Underwater Glider with Propellers

**Ngoc-Duc Nguyen** [1], **Hyeung-Sik Choi** [2,*] **and Sung-Wook Lee** [3]

1 Department of Electrical and Information Engineering, Seoul National University of Science and Technology, Seoul 01811, Korea; ducnn1908@gmail.com
2 Division of Mechanical Engineering, Korea Maritime and Ocean University, Busan 49112, Korea
3 Department of Naval Architecture and Ocean Engineering, Korea Maritime and Ocean University, Busan 49112, Korea; swlee@kmou.ac.kr
* Correspondence: hchoi@kmou.ac.kr; Tel.: +82-10-5581-2971

**Abstract:** This paper presents the modeling of a new ray-type hybrid underwater glider (RHUG) and an experimental approach used to robustly and adaptively control heading motion. The motions of the proposed RHUG are divided into vertical-plane motions and heading motion. Hydrodynamic coefficients in the vertical-plane dynamics are obtained using a computational fluid dynamics (CFD) method for various pitch angles. Due to the difficulty of obtaining accurate parameter values for the heading dynamics, a robust adaptive control algorithm was designed containing an adaptation law for the unknown parameters and robust action for minimizing environmental disturbances. For robust action against bounded disturbances, such as waves and ocean currents, sliding mode control was applied under the assumption that the bounds of the external disturbances are known. A direct adaptive algorithm for heading motion was applied in an experiment. Computer simulations of the proposed robust adaptive heading control are presented to demonstrate the robustness of the proposed control system in the presence of bounded disturbances. To verify the performance of the proposed controller for heading dynamics, several heading control experiments were conducted in a water tank and in the sea.

**Keywords:** robust adaptive control; vertical plane dynamics; heading dynamics; ray-type hybrid underwater glider

## 1. Introduction

Hybrid underwater gliders (HUG) have received much attention from underwater technology communities for oceanographic and military applications. For autonomous operation, the control design plays an important role in underwater vehicle development. A novel adaptive dynamic sliding mode control algorithm was developed for an autonomous underwater vehicle (AUV) [1]. This control system was found to be globally asymptotically stable based on Lyapunov theory, but no experimental verification was conducted. The nonlinearity of the rudder saturation was considered a disturbance when designing a sliding-mode-based adaptive control [2]. Experimental results of this control algorithm were provided using cross-type rudders. A robust generalized dynamics inversion control algorithm was presented with a model of an AUV, and only a numerical simulation was conducted to demonstrate the robustness of the proposed control design [3]. The upper bound on environmental disturbances was estimated using an adaptive tuning law in a novel adaptive second-order sliding mode control algorithm [4]. A robust finite-time trajectory tracking control algorithm was proposed with only simulation results [5]. Simulations and experimental results of an adaptive hybrid control algorithm were reported using the existing dynamical model of an underwater

glider (Petrel-II 200, developed by Tianjin University) [6]. Sliding mode tracking control for AUVs was proposed with a finite time disturbance observer, where asymptotic stability was ensured. However, only simulation results were reported, and this control algorithm was not experimentally verified [7]. An adaptive heading control was proposed for an underwater wave glider [8]. Model reference adaptive control (MRAC) and command governor adaptive control (CGAC) were compared in depth control experiments [9]. Faramin, M. et al. developed an adaptive sliding mode control for separated surge motion, an observer-based model adaptive control was applied for heading motion, and simulation results were presented to show the efficiency of the proposed method [10]. An adaptive chatter-free sliding mode control was proposed without the predefined bounds of external disturbances [11]. As described above, a number of studies were performed on adaptive and robust control algorithms, and performance was verified only through computer simulations.

In this study, a new ray-type hybrid underwater glider (RHUG) modeling was completed through CFD analysis, and its parameters for heading control were adapted through experiments in a water tank and in the sea. For the RHUG, hydrodynamic coefficients in the vertical-plane dynamics were obtained using a CFD method for various pitch angles. Due to the difficulty of obtaining accurate parameter values of the heading dynamics, a robust adaptive control algorithm was designed containing an adaptation law for the unknown parameters and robust action for minimizing environmental disturbances. Simulations and experiments were conducted to verify the proposed control algorithm.

The paper is organized as follows. The vertical-plane and heading dynamics are presented in Section 2. The stability of the proposed robust adaptive heading control is proved in Section 3. The robustness of the proposed algorithm is demonstrated through simulations in Section 4. Finally, the results of robust adaptive heading control in a water tank and at sea are presented in Section 5. The conclusions are outlined in Section 6.

## 2. RHUG Modeling

The six degrees of freedom (DOF) equations of motion of a fully submerged underwater vehicle, whose body axes coincide with the principal axes of inertia, can be written as [12]

$$
\begin{aligned}
m\left[\dot{u} - vr + wq - x_g\left(q^2 + r^2\right) + y_g\left(pq - \dot{r}\right) + z_g\left(pr + \dot{q}\right)\right] &= X \\
m\left[\dot{v} - wp + ur - y_g\left(r^2 + p^2\right) + z_g\left(qr - \dot{p}\right) + x_g\left(qp + \dot{r}\right)\right] &= Y \\
m\left[\dot{w} - wq + vp - z_g\left(p^2 + q^2\right) + x_g\left(rp - \dot{q}\right) + y_g\left(rq + \dot{p}\right)\right] &= Z \\
I_x\dot{p} + \left(I_z - I_y\right)qr + m\left[y_g\left(\dot{w} - uq + vp\right) - z_g\left(\dot{v} - wp + ur\right)\right] &= K \\
I_y\dot{q} + (I_x - I_z)rp + m\left[z_g\left(\dot{u} - vr + wq\right) - x_g\left(\dot{w} - uq + vp\right)\right] &= M \\
I_z\dot{r} + \left(I_y - I_x\right)pq + m\left[x_g\left(\dot{v} - wp + ur\right) - y_g\left(\dot{u} - vr + wq\right)\right] &= N
\end{aligned}
\tag{1}
$$

where $u, v,$ and $w$ are the linear velocities of the origin $O$ of the body-fixed frame; $p, q,$ and $r$ are the angular velocities in the body-fixed frame; $\phi, \theta,$ and $\psi$ are Euler angles in the earth-fixed frame; $x_g, y_g,$ and $z_g$ are the position of the center of gravity (CG; Figure 1) in the moving frame $Ox_0y_0z_0$ ; $X, Y,$ and $Z$ are the forces acting on the vehicle in the body-fixed frame; and $K, M,$ and $N$ are the moments acting on the vehicle in the body-fixed frame. The kinematic system can be driven by Euler angles as shown in Equation (2):

$$
\begin{aligned}
\dot{x} &= uc\psi c\theta + v(c\psi s\theta s\phi - s\psi c\phi) + w(s\psi s\phi + c\psi c\phi s\theta) \\
\dot{y} &= us\psi c\theta + v(c\psi c\phi - s\phi s\theta s\psi) + w(s\theta s\psi c\phi - c\psi s\phi) \\
\dot{z} &= -us\theta + vc\theta s\phi + wc\theta c\phi \\
\dot{\phi} &= p + qs\phi t\theta + rc\phi t\theta \\
\dot{\theta} &= qc\phi - rs\phi \\
\dot{\psi} &= q\frac{s\phi}{c\theta} + r\frac{c\phi}{c\theta}
\end{aligned}
\tag{2}
$$

and if $j \in \{\phi; \theta; \psi\}$, then $cj$ is $\cos(j)$; $sj$ is $\sin(j)$; and $tj$ is $\tan(j)$.

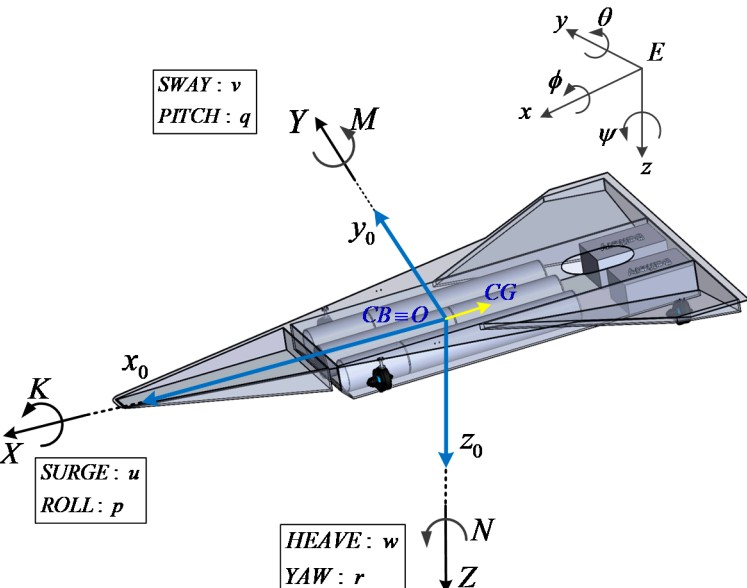

**Figure 1.** Coordinate system of ray-type hybrid underwater glider (RHUG). $O_{x_0 y_0 z_0}$ is the body-fixed frame, and $E_{xyz}$ is the earth-fixed coordinate system.

The external force and moment vector contain three components: $[X, Y, Z, K, M, N]^T = \tau_H + \tau + \tau_e$. The hydrodynamic forces and moments, $\tau_H$, can be estimated using CFD. The control input $\tau$ is generated by thrusters, a moving mass (Figure 2a) and buoyancy engines (Figure 2b). Finally, the environmental input $\tau_e$ is the disturbance from ocean currents and waves.

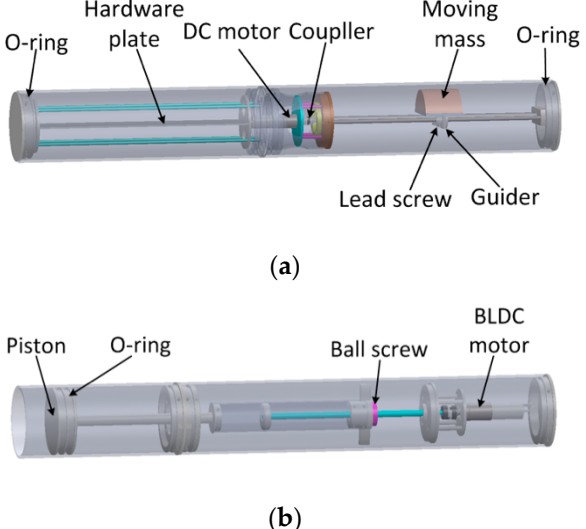

**(a)**

**(b)**

**Figure 2.** Mass-shifter (**a**) and buoyancy engine (**b**) design. A DC motor is used in the mass-shifter, and a Brushless DC motor is installed in the buoyancy engine.

In the RHUG design described in Figure 3 and Table 1, the sway and roll dynamics do not have any actuators. For now, the stability of the roll motion is dependent on the vertical passive stabilizer, as shown by the CFD analysis result in Figure 4. As the main objective of the RHUG hull design is gliding motion, the sway and roll dynamics were neglected in the RHUG modeling in this study. Under this condition, the dynamics of the RHUG are separated into two, as shown in Figure 5. The first dynamics are surge–heave–pitch motion, which are the vertical-plane dynamics in the *Exz* plane. In the vertical-plane motion, the coupling terms among surge, heave, and pitch motions cannot be neglected in the RHUG gliding motion. The second type dynamics is yaw motion, which is

heading motion. Therefore, the RHUG dynamics are presented in terms of vertical motion and heading motion individually.

**Table 1.** Specification of RHUG.

| Parameter | Value |
|---|---|
| Length × Width × Height | 2.48 m × 1.8 m × 0.22 m |
| Static mass | 115 kg |
| Moving mass | 2 kg |
| Net buoyancy | ±1.4 kg |

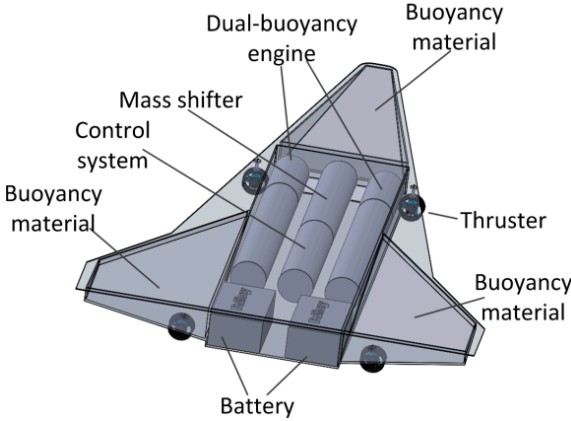

**Figure 3.** RHUG design.

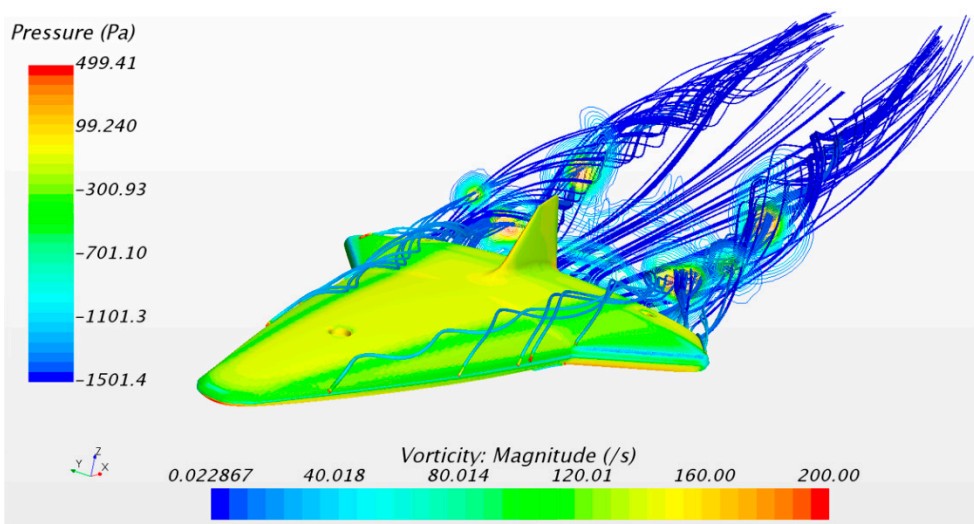

**Figure 4.** Vertical static drift test for varying pitch angle.

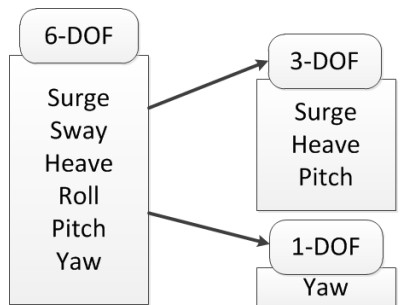

**Figure 5.** Modeling concept. DOF: degree of freedom.

The planar motion mechanism (PMM) test for the hydrodynamic coefficients is very expensive. Therefore, to obtain the hydrodynamic coefficients of this new hull design, CFD calculation is performed. One of the results of the vertical drift calculation that varies from −15° to 15° is presented in Figure 4 as an example. In Figure 4, the surface contour on the RHUG hull indicates the pressure field acting on the hull. The line contour around the hull is the fluid flow in the vertical drift test. From the CFD calculations, we found that the weak vortex shedding occurred at the both wing edges. The surge and heave force acting on the RHUG hull in the various vertical drift angles are presented in Figures 6 and 7, respectively.

The acting force in the surge direction was computed from pitch angles of −15° to 15°. In the drift calculation, the nondimensional maximum and minimum surge force are 0.0048 at −5° and −0.0039 at −15°, respectively, as shown in Figure 6. The heave force acting on the hull was also calculated from a pitch angle of −15° to 15°. The magnitude of the nondimensional heave force increases as the magnitude of the pitch angle increases, as shown in Figure 7. The dimensionless hydrodynamic coefficients from the CFD results are shown in Table 2.

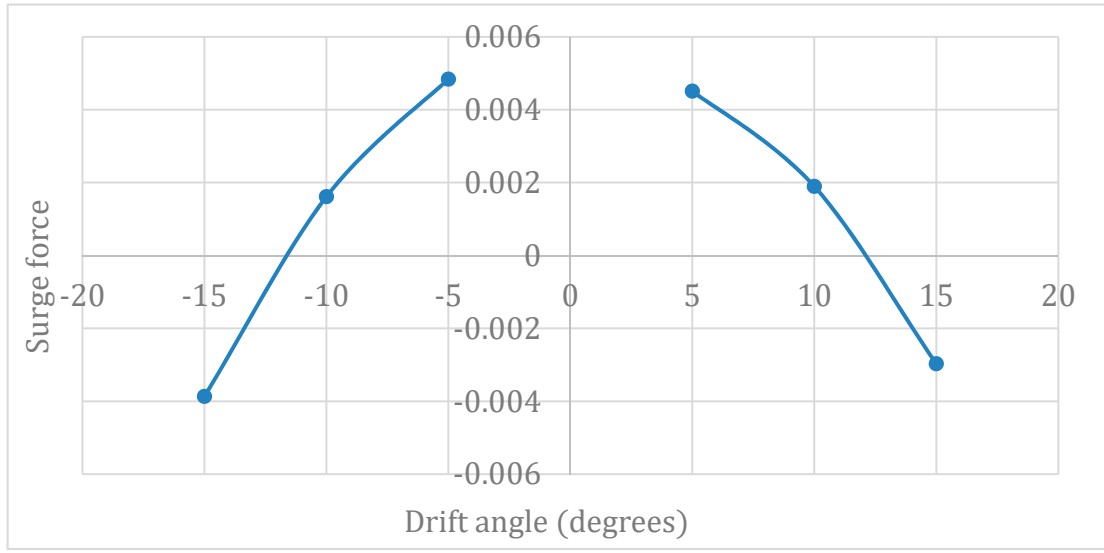

**Figure 6.** Vertical static drift calculation result for surge motion.

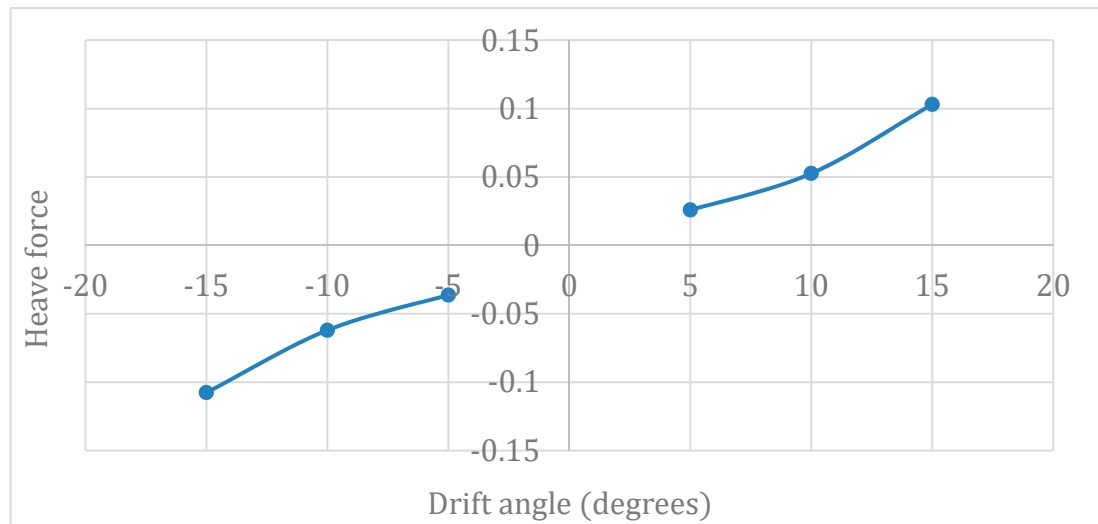

**Figure 7.** Vertical static drift calculation result for heave motion.

**Table 2.** Dimensionless hydrodynamics coefficients (CFD method).

| Parameter | Value | Parameter | Value | Parameter | Value |
|---|---|---|---|---|---|
| $X_{\dot{u}}$ | −0.03 | $Z_{\dot{w}}$ | −0.011836 | $M_{\dot{w}}$ | −0.022352 |
| $X_{uu}$ | −0.0063 | $Z_{\dot{q}}$ | −0.004774 | $M_{\dot{q}}$ | −0.003823 |
| $X_{uw}$ | 0.1485 | $Z_{uu}$ | −0.0052 | $M_{uu}$ | −0.0018 |
| $X_{ww}$ | 0.0013 | $Z_{uw}$ | −0.3204 | $M_{uw}$ | 0.0117 |
| - | - | $Z_{ww}$ | 0.0356 | $M_{ww}$ | −0.0173 |
| - | - | $Z_{www}$ | −1.623 | $M_{www}$ | 0.6989 |

Here, $X_{\dot{u}}$ is the added mass coefficient in the surge motion; $Z_{\dot{w}}$ and $Z_{\dot{q}}$ are the added mass coefficients in the heave motion; $M_{\dot{w}}$ and $M_{\dot{q}}$ are the added mass coefficients in the pitch motion; $X_{uu}$, $X_{uw}$, and $X_{ww}$ are the hydrodynamic coefficients in the surge motion; $Z_{uu}$, $Z_{uw}$, $Z_{ww}$, and $Z_{www}$ are the hydrodynamic coefficients in the heave motion; and $M_{uu}$, $M_{uw}$, $M_{ww}$, and $M_{www}$ are the hydrodynamic coefficients in the pitch motion [12].

By reducing the 6-DOF dynamics system, the vertical system, which is important for gliding motion, is described in Equation (3) using the resulting hydrodynamic coefficients in Table 2.

$$
\begin{aligned}
\dot{x} &= u\cos\theta + w\sin\theta \\
\dot{z} &= -u\sin\theta + w\cos\theta \\
\dot{\theta} &= q \\
\left(m - X_{\dot{u}}\right)\dot{u} &= -mz_g\dot{q} + mx_g q^2 - mwq + Z_{\dot{w}}wq + X_{uu}u^2 + X_{ww}w^2 \\
&\quad + X_{uw}uw - (W - B)\sin\theta + \tau_w\sin\theta + \tau_u + \tau_{eu} \\
\left(m - Z_{\dot{w}}\right)\dot{w} &= \left(mx_g + Z_{\dot{q}}\right)\dot{q} + mz_g q^2 + muq - X_{\dot{u}}uq + Z_{uu}u^2 + Z_{uw}uw + Z_{www}w^3 \\
&\quad + Z_{ww}w^2 + (W - B)\cos\theta + \tau_w\cos\theta + \tau_{ew} \\
\left(I_{yy} - M_{\dot{q}}\right)\dot{q} &= -mz_g\dot{u} + \left(mx_g - M_{\dot{w}}\right)\dot{w} - mz_g qw - Z_{\dot{w}}wu - Z_{\dot{q}}qu + X_{\dot{u}}uw + M_{uu}u^2 \\
&\quad + M_{uw}uw + M_{ww}w^2 + M_{www}w^3 - \left(z_g W - z_b B\right)\sin\theta \\
&\quad - \left(x_g - x_b B\right)\cos\theta + \tau_q + \tau_{eq}
\end{aligned}
\tag{3}
$$

where $W$ is the vehicle weight; $B$ is the buoyancy force; $x_b$ and $z_b$ are the coordinates of the center of buoyancy; $\tau_u$, $\tau_w$ and $\tau_q$ are the control inputs induced by the thrusters, buoyancy engines, and moving

mass, respectively; $\tau_{eu}$, $\tau_{ew}$, and $\tau_{eq}$ are the environmental disturbances in the surge, heave, and pitch motions, respectively. The decoupled yaw dynamics of the underwater glider can be written as

$$\dot{\psi} = r$$
$$\left(I_{zz} - N_{\dot{r}}\right)\dot{r} = N_r r + N_{|r|r}|r|r + \tau_r + d \tag{4}$$

where $r$ is the yaw rate; $\psi$ is the heading angle; $I_{zz}$ is the moment of inertia about the $Oz_0$ axis; $N_{\dot{r}}$ is the added mass coefficient; $N_r$ and $N_{|r|r}$ are the linear and quadratic damping coefficients, respectively; $\tau_r$ is the torque of thrusters; and $d$ is the external disturbance induced by currents and waves.

$$T_i = \begin{cases} 0.68u_t - 4.795 & 7 < u_t \leq 80 \\ 0.54u_t + 3.836 & -80 \leq u_t < -7 \\ 0 & -7 \leq u_t \leq 7 \end{cases} \tag{5}$$

The thrusters in this platform were T200 thrusters (Blue Robotics, Torrance, CA). The experimental data were provided by Blue Robotics. In Figure 8, the thrust force $T_i$ ranges from −40 to 50 N with the input signal $u_t$ ranging from −80 to 80%. This relationship can be illustrated as the set of equations in Equation (5), where the subscript letter $i \in \{1, 2, 3, 4\}$. The control force and moment can be calculated from thruster force using Equations (6) and (7), where $\tau_u$ is the surge control input for speed control in the body-fixed frame, $\tau_r$ is the yaw moment in the body-fixed frame, $T_1$ and $T_2$ are the forces of thrusters on the starboard, $T_3$ and $T_4$ are the forces of thrusters on the stern, $d_1$ is the distance of the two thrusters on the starboard, and $d_2$ is the distance of two thrusters in the stern:

$$\tau_u = T_1 + T_2 + T_3 + T_4 \tag{6}$$

$$\tau_r = (T_1 - T_2)\frac{d_1}{2} + (T_3 - T_4)\frac{d_2}{2} \tag{7}$$

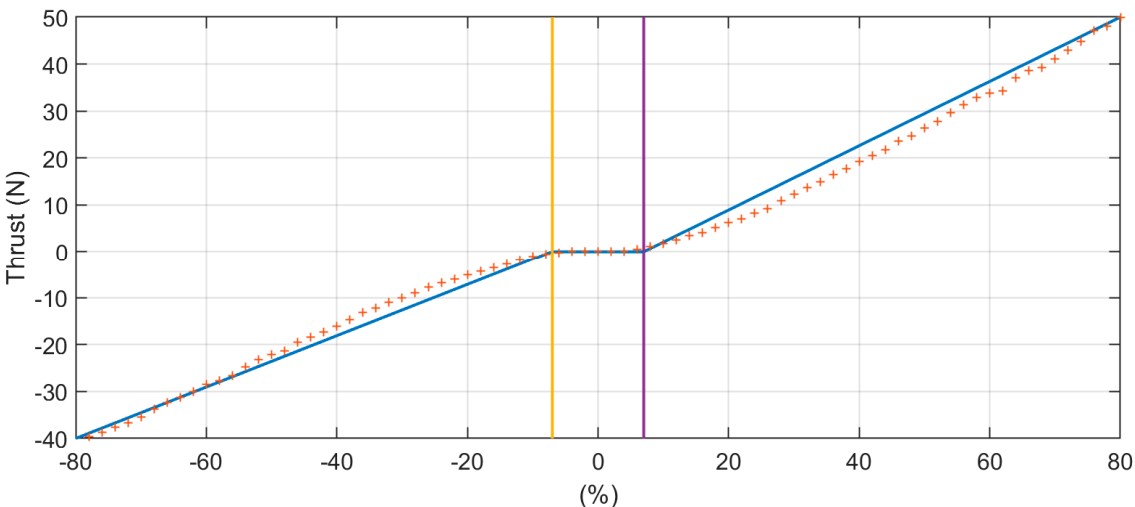

**Figure 8.** Thruster force vs. percentage input.

## 3. Robust Adaptive Control

The heading error can be defined as

$$e_1 = \psi - \psi_d \tag{8}$$

where $\psi_d$ is the reference of the heading control. The heading error dynamics can be obtained by taking the derivative of Equation (8):

$$\dot{e}_1 = \dot{\psi} - \dot{\psi}_d \tag{9}$$

By substituting Equation (4) into Equation (9), we obtain

$$\dot{e}_1 = r - \dot{\psi}_d = e_2 - \dot{\psi}_d + r_d \tag{10}$$

where $r$ is the yaw rate measurement, $\dot{\psi}_d$ is the desired yaw rate, $r_d$ is the virtual control input introduced to stabilize the heading error dynamics, and $e_2$ is the error between the yaw rate and the virtual control input in Equation (11).

$$e_2 = r - r_d \tag{11}$$

Based on the heading error dynamics in Equation (10), the virtual control can be designed as

$$r_d = -k_1 e_1 + \dot{\psi}_d \tag{12}$$

where $k_1$ is the positive control gain. The magnitude of the gain $k_1$ affects the convergence rate of the system in Equation (13). If $e_2$ tends to 0 while the time $t$ tends to $+\infty$ , then the system in Equation (13) will be asymptotically stable.

$$\dot{e}_1 = -k_1 e_1 + e_2 \tag{13}$$

The derivative of $e_2$ should be obtained for further stability analysis. By differentiating Equation (11), we obtain the derivative of $e_2$ :

$$\dot{e}_2 = \dot{r} - \dot{r}_d \tag{14}$$

The control input $\tau_r$ can be seen in $\dot{e}_2$ by substituting Equation (4) into Equation (14), and the final derivative of $e_2$ is

$$\dot{e}_2 = \frac{1}{a}(br + c|r|r + \tau_r + d) - \dot{r}_d \tag{15}$$

where $a = I_{zz} - N_{\dot{r}}$ is the positive inertia term, $b = N_r$ , $c = N_{|r|r}$ , and the derivative of the virtual control is

$$\dot{r}_d = \ddot{\psi}_d + k_1^2 e_1 - k_1 e_2 \tag{16}$$

where $\ddot{\psi}_d$ is the desired yaw acceleration. The desired yaw rate $\dot{\psi}_d$ and the desired yaw acceleration $\ddot{\psi}_d$ are normally assigned as zero.

The sliding surface function is

$$s = e_2 + \lambda e_1$$
$$s = e_2 + \lambda e_1 \tag{17}$$

where $\lambda$ is the positive weight constant between $e_1$ and $e_2$. To reduce the chattering issue in sliding mode control, the new variable is defined as

$$s_\Delta = s - \phi sat\left(\frac{s}{\phi}\right) \tag{18}$$

where

$$\begin{cases} sat\left(\frac{s}{\phi}\right) = \frac{s}{\phi} & if \left|\frac{s}{\phi}\right| \le 1 \\ sat\left(\frac{s}{\phi}\right) = sgn\left(\frac{s}{\phi}\right) & otherwise \end{cases} \tag{19}$$

and $\phi$ is the constant boundary layer of the saturation function. The inertia term $a$ is always positive; then, the Lyapunov candidate can be designed as

$$V_2 = \frac{1}{2}as_\Delta^2 + \frac{1}{2}\tilde{a}_2^T P_2^{-1}\tilde{a}_2 \tag{20}$$

where $P_2 \in R^{3 \times 3}$ is the positive constant matrix, $\tilde{a}_2 = \hat{a}_2 - a_2$ is the estimate error of parameter vector $a_2$, and $\hat{a}_2$ is the estimated value of $a_2$. The constant parameter vector $a_2$ is derived later in the time derivative of the Lyapunov function, which can be derived as

$$\dot{V}_2 = as_\Delta\left(\dot{e}_2 + \lambda\dot{e}_1\right) + \dot{\hat{a}}_2^T P_2^{-1}\tilde{a}_2 \tag{21}$$

By substituting Equations (13) and (15) into Equation (21), we obtain

$$\dot{V}_2 = s_\Delta\left(br + cr|r| + \tau_r + d - a\dot{r}_d + a\lambda\dot{e}_1\right) + \dot{\hat{a}}_2^T P_2^{-1}\tilde{a}_2. \tag{22}$$

To design the adaptation law for $\hat{a}_2$ , the unknown term should be described as

$$Y_2 a_2 = br + cr|r| - a\left(\dot{r}_d - \lambda\dot{e}_1\right) \tag{23}$$

where $Y_2 = \begin{bmatrix} r & r|r| & -\dot{r}_d + \lambda\dot{e}_1 \end{bmatrix}$ is the regressive vector of state variables and reference. In other words, $Y_2$ can be determined by the heading angle $\psi$, yaw rate $r$, desired heading angle $\psi_d$, desired yaw rate $\dot{\psi}_d$, and desired yaw acceleration $\ddot{\psi}_d$. Accordingly, the parameter vector $a_2$ is $\begin{bmatrix} b & c & a \end{bmatrix}^T$. Then, the derivative of $V_2$ can be rewritten as

$$\dot{V}_2 = s_\Delta(Y_2 a_2 + d + \tau_r) + \dot{\hat{a}}_2^T P_2^{-1}\tilde{a}_2 \tag{24}$$

The actual control input $\tau_r$ can be designed using Equation (24):

$$\tau_r = -Y_2\hat{a}_2 - \hat{d} - k_2 s_\Delta - k_3 sat\left(\frac{s}{\phi}\right) \tag{25}$$

where $\hat{d}$ is the mean value of the wave-form disturbance, $k_2$ is the positive stabilizing gain, and $k_3$ is the positive gain in sliding mode control. By applying this control law into Equation (24), $\dot{V}_2$ is rewritten as

$$\dot{V}_2 = -k_2 s_\Delta^2 + s_\Delta Y_2\tilde{a}_2 + \left(d - \hat{d}\right)s_\Delta - k_3 sat\left(\frac{s}{\phi}\right)s_\Delta + \dot{\hat{a}}_2^T P_2^{-1}\tilde{a}_2 \tag{26}$$

By canceling the group of $s_\Delta Y_2\tilde{a}_2$ and $\dot{\hat{a}}_2^T P_2^{-1}\tilde{a}_2$, the adaptation law can be obtained:

$$\dot{\hat{a}}_2 = -P_2^T Y_2^T s_\Delta \tag{27}$$

Then, with the above adaptation, the derivative of the Lyapunov function is derived as

$$\dot{V}_2 = -k_2 s_\Delta^2 + \left(d - \hat{d}\right)s_\Delta - k_3 sat\left(\frac{s}{\phi}\right)s_\Delta \quad \left(\dot{V}_2 = 0 \; as \; s = 0\right) \tag{28}$$

$\left(d - \hat{d}\right) \le \left|d - \hat{d}\right|$ , $s_\Delta \le |s_\Delta|$ and $sat\left(\frac{s}{\phi}\right) \le 1$ . Therefore, an inequality can be derived in Equation (29) from Equation (28).

$$\dot{V}_2 \le -k_2 s_\Delta^2 + \left|d - \hat{d}\right||s_\Delta| - k_3 sat\left(\frac{s}{\phi}\right)|s_\Delta| \tag{29}$$

Due to the unknown nominal value of the parameters, $\hat{d}$ cannot be estimated using nonlinear observers. Then, the estimation of ocean currents and waves can be assumed to be equal to zero, and the bounds of disturbances can be used as the tuning gain in different environments. This assumption is reasonable because the current and wave disturbances can be modeled as a sinusoidal function with zero mean ( $\hat{d} = 0$). Therefore, $\dot{V}_2$ is negative definite as described in Equation (30), if and only if

$k_3 \geq |d - \hat{d}|$. Then, $k_3$ can be experimentally tuned as the maximum of environmental moment acting on the vehicle.

$$\dot{V}_2 \leq -k_2 s_\Delta^2 \leq 0 \tag{30}$$

Using Barbalat's lemma, $V_2$ is positive definite and $\dot{V}_2$ is negative definite, then $\dot{V}_2 \to 0$ as $t \to +\infty$. This implies that $s_\Delta \to 0$ as $t \to +\infty$ or $s \leq \phi$ as $t \to +\infty$. In other words, with the control law defined in Equation (25), the sliding quantity $s$. approaches and stays within the constant boundary layer $\phi$.

Notably, Equation (30) does not guarantee that the estimation error of the parameters converge to zero. The parameter estimates can be viewed as extra states of the controller that help with the control task.

## 4. Simulation

For simulation purposes, the RHUG heading dynamics were assumed to have the parameters listed in Table 3. The external disturbance was chosen as $5sin\left(\frac{\pi}{2t}\right)$; then, $|d - \hat{d}| \leq 5$. Nm and $\hat{d} = 0$. The input saturation was defined as $|\tau_r| \leq 20$ Nm. The heading angle was regulated between $\psi_d = -30°$ and $30°$. The desired yaw rate and yaw acceleration were chosen as $\dot{\psi}_d = 0$ degrees/s and $\ddot{\psi}_d = 0$ degrees/s². Other gains were chosen as $k_1 = 1$, $k_2 = 30$, and $\lambda = 1$.

**Table 3.** Parameters for heading dynamics in simulation.

| Parameter | Value |
|:---:|:---:|
| $a$ | 13.1613 |
| $b$ | −18.7246 |
| $c$ | −3.5488 |

### 4.1. Adaptive Heading Control

This subsection describes the effect of external disturbance on direct adaptive control. In this simulation, the control law is constructed as

$$\tau_r = -Y_2 \hat{a}_2 - \hat{d} - k_2 s_\Delta. \tag{31}$$

The heading control without robust action cannot mitigate the effects of external disturbances. The external disturbance causes significant error in heading regulation, as shown in Figure 9a. The yaw rate cannot converge to zero, as shown in Figure 9b. In Figure 9a, the straight red line is the desired heading angle and the wavy blue line is the actual heading angle. In Figure 9b, the yaw rate is the blue line and the virtual control input is the red line.

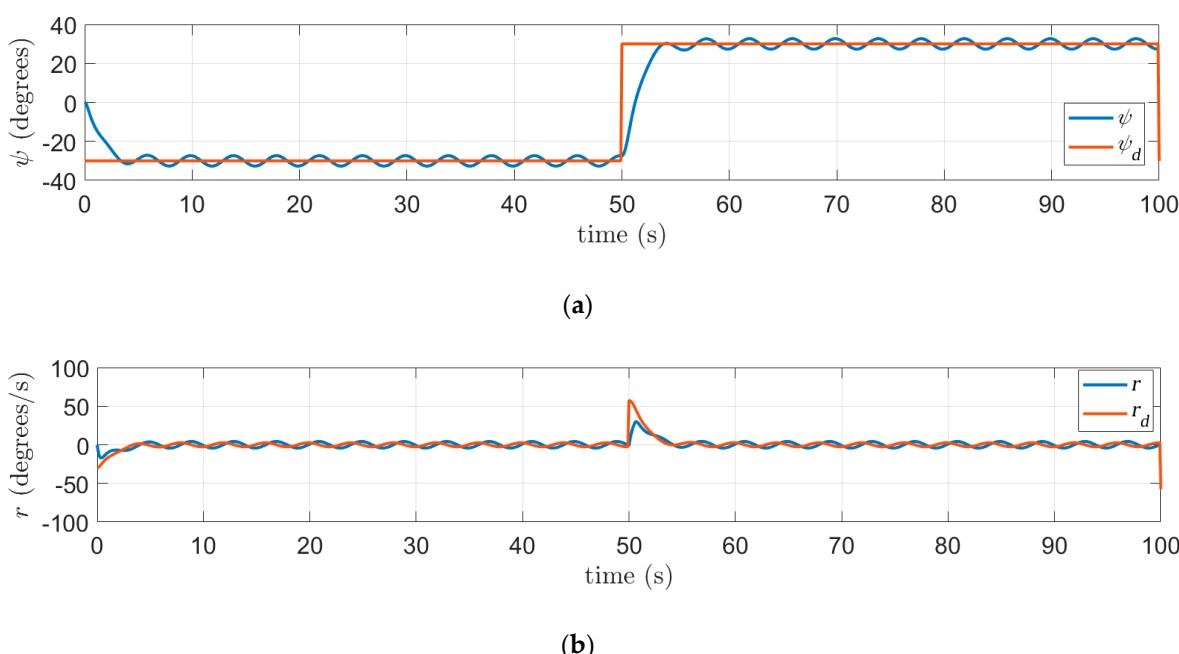

**(a)**

**(b)**

**Figure 9.** Simulation results of heading control performance without robust control. (**a**) Heading angle is regulated to −30° and 30°. (**b**) Yaw rate is regulated at 0 degrees/s.

Figure 10 shows the control effort of the adaptive control in Equation (31). This control smoothes the motion, but cannot eliminate the external disturbances. The adaptation law for the unknown parameters does not converge to any constant values, as shown in Figure 11, whereas the real parameters are $a = 13.1613$, $b = -18.7246$, and $c = -3.5488$. Due to the effects of external disturbances, the estimation of the three parameters of the heading dynamics cannot converge to a constant value.

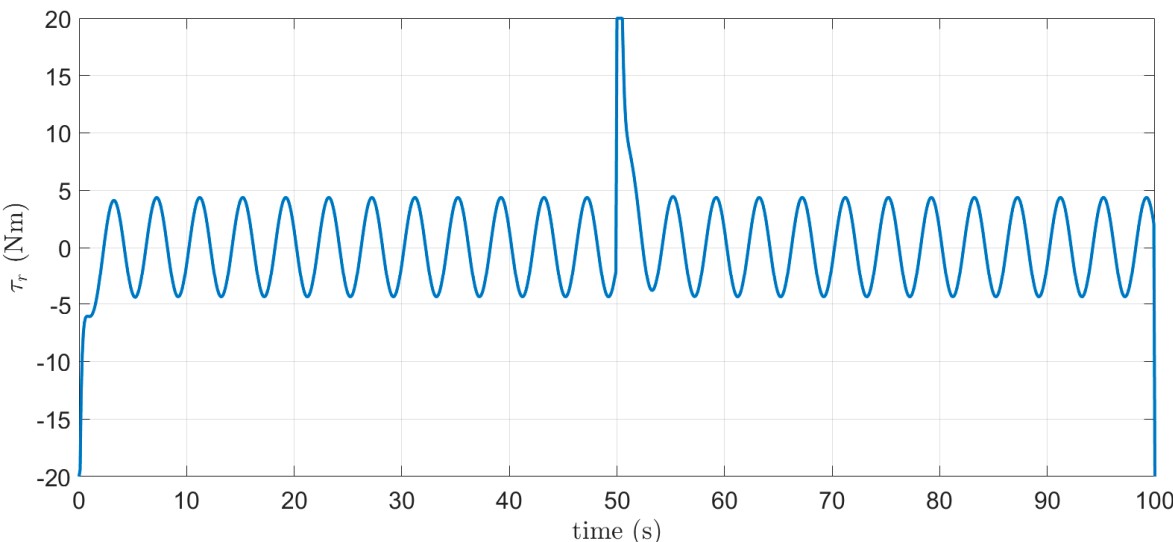

**Figure 10.** Simulation result of the control input of the direct adaptive control without robust action.

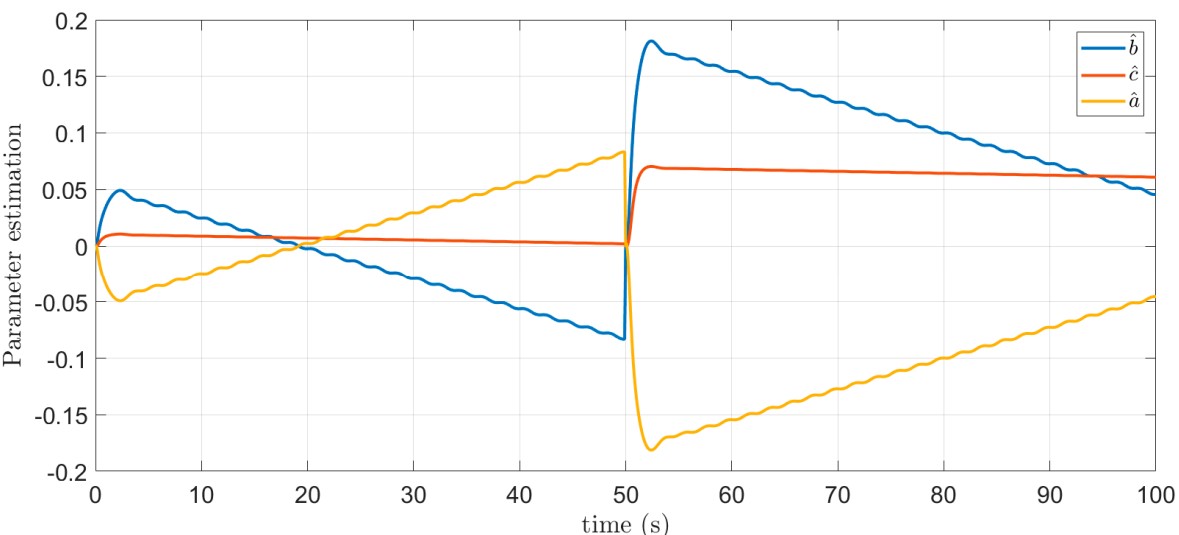

**Figure 11.** Simulation results of parameter adaptation without robust action.

*4.2. Robust Adaptive Heading Control*

Under the same conditions of the simulation of applying adaptive heading control in the previous subsection, the robust adaptive control was used as described in Equation (25). The constant boundary was chosen as $\phi = 0.04$.

The performance of the robust adaptive heading control is shown in Figure 12. In the presence of external disturbances, the heading angle is regulated well along the desired heading angle, as shown in Figure 12a. The yaw rate also converges to zero following the virtual input in Figure 12b. Due to the small boundary layer $\phi$, the chattering appears quickly at the beginning of the regulation task. In Figure 13, the control input has similar smoothness to Figure 10, but this input can minimize the external disturbance well.

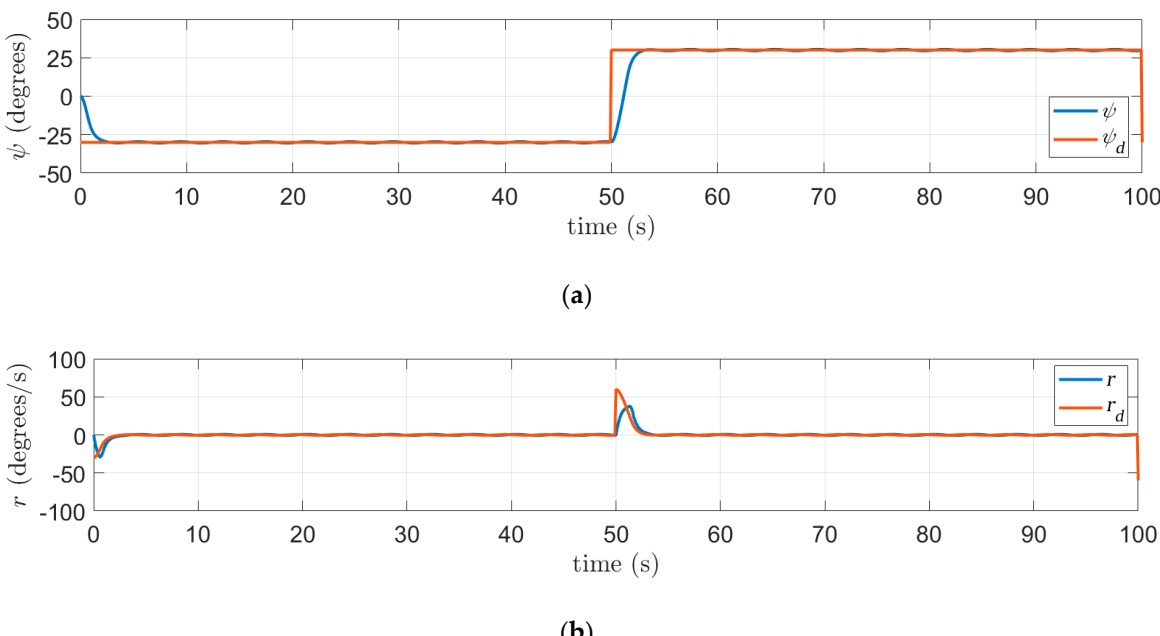

(**a**)

(**b**)

**Figure 12.** Simulation results of heading performance with robust control. (**a**) Heading angle is regulated to $-30°$ and $30°$. (**b**) Yaw rate is regulated at $0$ degrees/s.

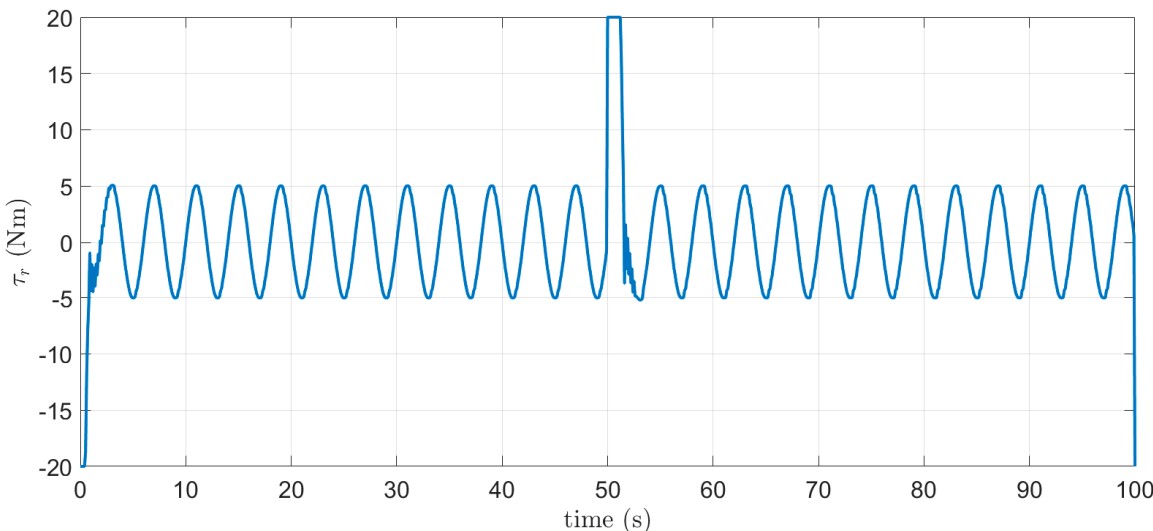

**Figure 13.** Simulation results of the control input of robust adaptive heading control.

In the parameter adaptation, convergence can be observed in Figure 14. All three estimated parameters converge to three constant values in each regulation task, and the real parameters are $a = 13.1613$, $b = -18.7246$, and $c = -3.5488$. The estimation of adaptive control does not converge to real constant parameters. This also shows that once the system enters the boundary layer, the adaptation action decreases, as the error dynamics do not excite the adaptation action and only robust control is used to eliminate the effect of external disturbance.

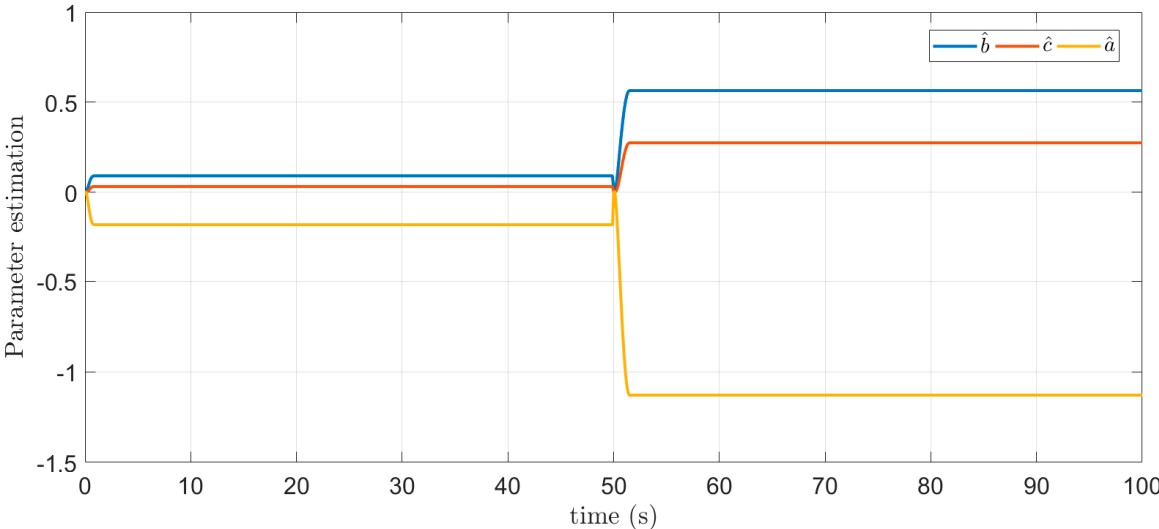

**Figure 14.** Simulation behavior of parameter adaptation.

## 5. Experimental Results

### 5.1. Heading Control in the Water Tank

We performed heading control in a water tank using robust adaptive control. The attitude heading reference system (AHRS), named the MTi model from XSENS (Enschede, The Netherlands), was used to measure the heading angle, and the yaw rate of the vehicle was measured using the gyro sensor inside the AHRS. Figure 15 shows the result of heading control in the water tank using robust adaptive control. The heading angle was regulated to 60° with less than 0.5° error, and the virtual control input and the yaw rate were regulated at zero, as shown in Figure 15a,b, respectively.

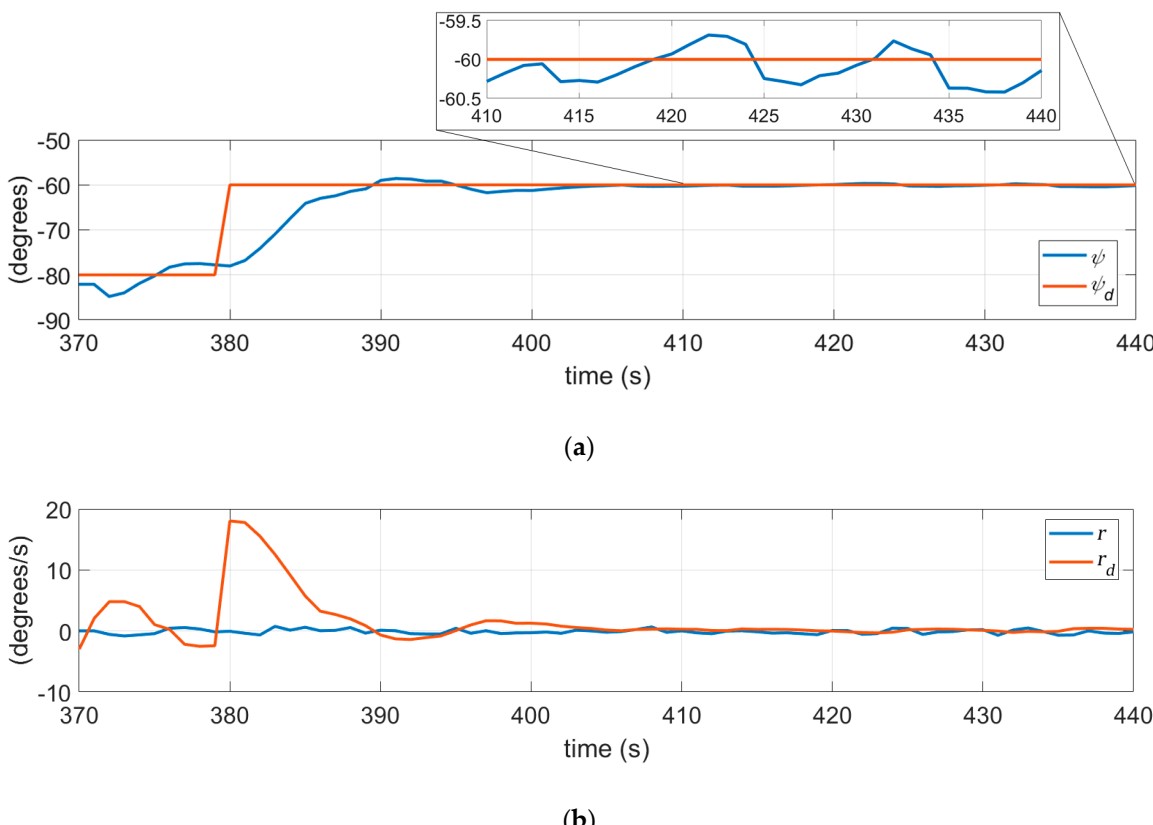

**Figure 15.** Experimental results of robust adaptive heading control in the water tank. (**a**) Heading control performance and (**b**) virtual control performance.

The robust adaptive heading control input is shown in Figure 16. The three parameters were estimated using the adaptation algorithm in the experiment, as shown in Figure 17. Convergence of the parameter adaptation was not achieved due to the noise from the yaw rate signal, which did not converge to zero as shown in Figure 15b. However, the tendency of the adaptation in the water tank was similar to the simulation in Figure 14. The final values of $\hat{b}$ and $\hat{c}$ had opposite signs of $\hat{a}$ in both the simulation and experiment.

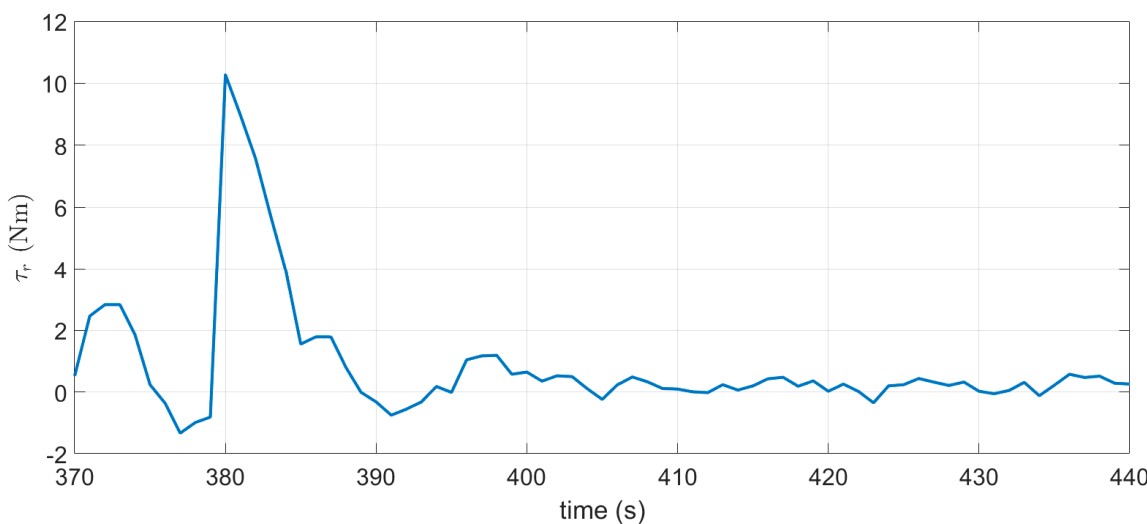

**Figure 16.** Experimental control input of robust adaptive control in the water tank.

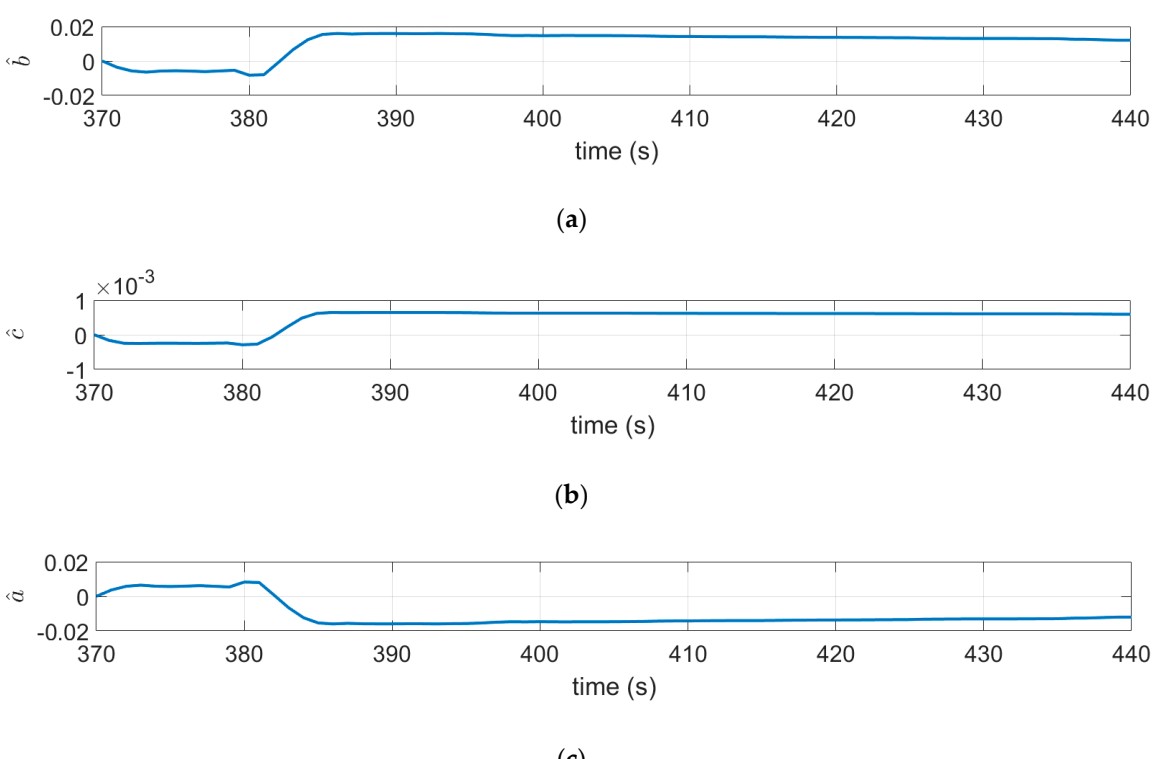

**Figure 17.** Experimental adaptation of three estimations: (**a**) estimation of parameter *b*, (**b**) estimation of parameter *c*, and (**c**) estimation of parameter *a*.

## 5.2. Heading Control in the Sea Test

One gliding test cycle with robust adaptive heading control was conducted in front of Korea Maritime and Ocean University (KMOU, Busan, South Korea) as shown in Figure 18. Figure 18a–d describes the descending motion of RHUG at a −90° heading angle, whereas Figure 18e–h shows the ascending motion of the RHUG at a −90° heading angle. The assumption for the external disturbances is the same as for the simulation with the disturbance bound of 5 Nm, as they could not be measured. All control gains were the same as in Section 4.

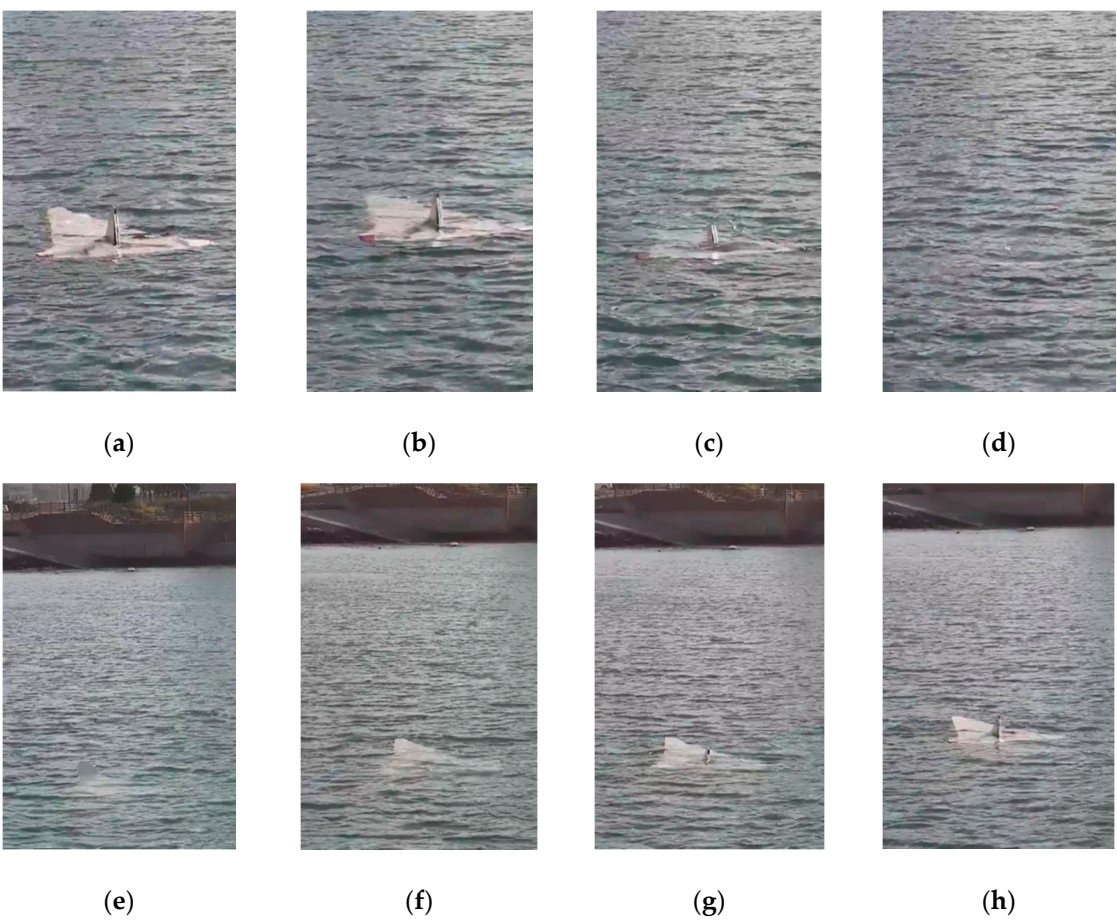

**Figure 18.** One cycle of gliding test in the sea: (**a**–**d**) descending motion and (**e**–**h**) ascending motion.

Figure 19 shows the heading performance when using robust adaptive control during the sea trial. The heading angle was regulated at −90° with an error of less than 4°. The yaw rate and virtual control input converged to around zero at the end of the experiment, which is good convergence. The control input of the robust adaptive heading control in the sea is shown in Figure 20.

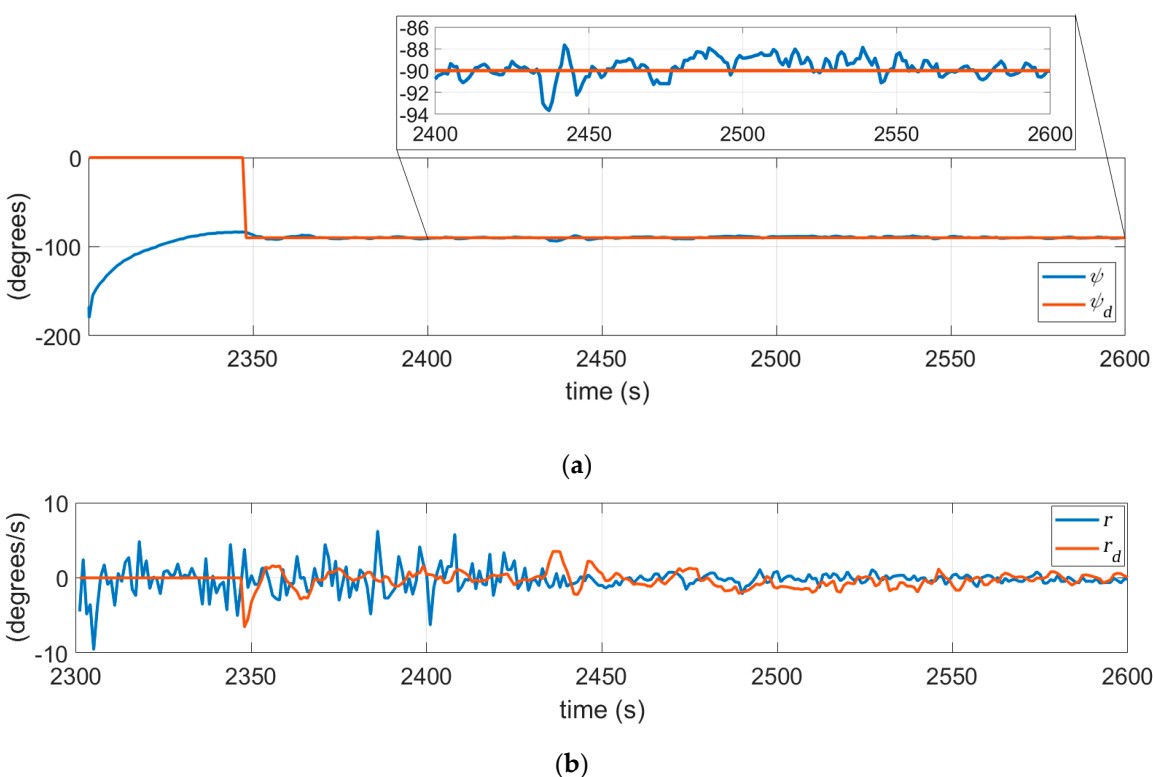

(**a**)

(**b**)

**Figure 19.** Experimental performance of heading and yaw rate in the sea in front of the Korea Maritime and Ocean University (KMOU, Busan, South Korea): (**a**) Heading control performance (**b**) virtual control performance.

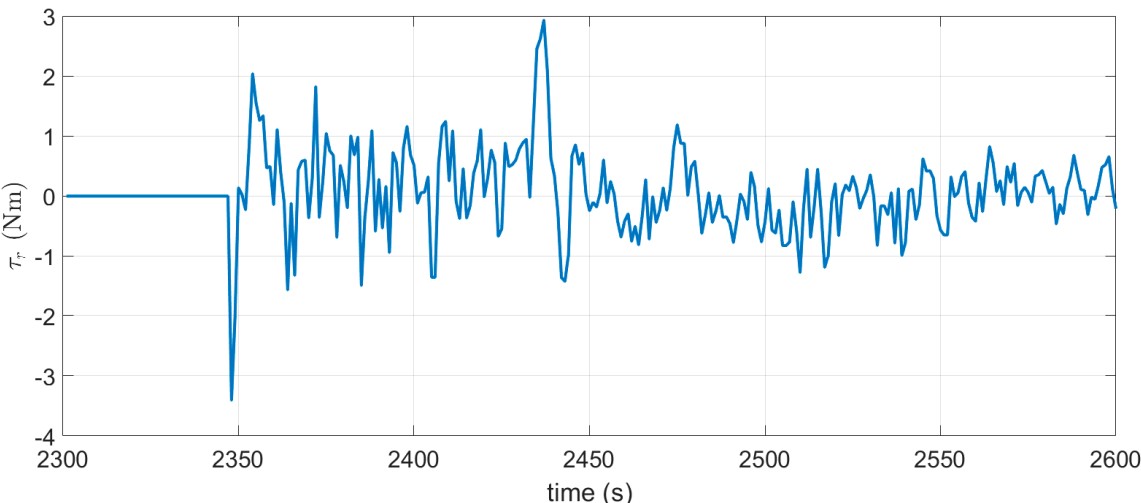

**Figure 20.** Control input of robust adaptive control during the gliding test in the sea.

The parameter adaptation showed the same tendency as the simulation. The signs of $\hat{b}$ and $\hat{c}$ were opposite to that of $\hat{a}$, as shown in Figure 21. The convergence of the constant parameter estimation was poor due to the non-zero convergence of the yaw rate signal. The estimated parameters $\hat{a}$, $\hat{b}$, and $\hat{c}$ could not be guaranteed to converge to the real parameters using robust adaptive control.

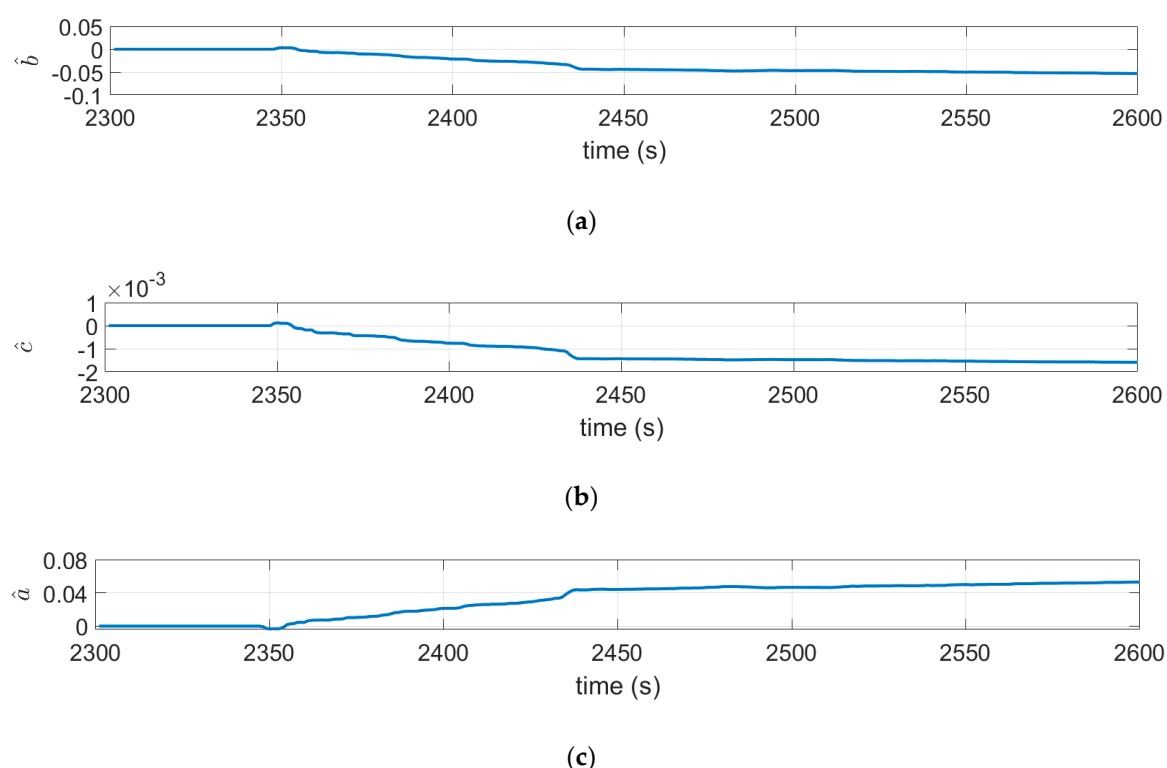

**Figure 21.** Experimental adaptation of three unknown parameters during the gliding test in the sea: (**a**) estimation of parameter $b$, (**b**) estimation of parameter $c$, (**c**) estimation of parameter $a$.

## 6. Conclusions

This paper presented the modeling of a new ray-type hybrid underwater glider and the experimental results when using robust adaptive control for heading motion. Hydrodynamic coefficients of vertical-plane dynamics were obtained through vertical static drift test using the CFD method with different pitch angles. During heading control, as the system parameters were unknown, a robust adaptive control algorithm was proposed without knowledge of the nominal value of the unknown parameters. The simulations verified the good performance of the proposed algorithm despite the unknown parameters and external disturbances. In the experiments, the bounds of the disturbances were tuned to mitigate the effects of environmental disturbances. Both simulations and experiments had good convergence in yaw rate and virtual control input. In the water tank experiment, the heading error was less than $0.5°$, whereas during the sea experiment, the heading error was less than $4°$.

**Author Contributions:** Conceptualization, N.-D.N., S.-W.L. and H.-S.C.; methodology, N.-D.N., S.-W.L. and H.-S.C.; software, N.-D.N. and S.-W.L.; validation, N.-D.N. and H.-S.C.; formal analysis, N.-D.N.; investigation, N.-D.N.; resources, N.-D.N., S.-W.L. and H.-S.C.; data curation, N.-D.N. and S.-W.L.; writing—original draft preparation, N.-D.N.; writing—review and editing, N.-D.N., S.-W.L. and H.-S.C.; visualization, N.-D.N.; supervision, H.-S.C.; project administration, H.-S.C.; funding acquisition, H.-S.C.

**Funding:** This research received no external funding.

**Acknowledgments:** This is results of the research project entitled "Data Collection System with Underwater Glider" (19-SN-MU-01) and is a part of the result of the research project (18-SN-RB-01).

**Conflicts of Interest:** The authors declare no conflict of interest.

## List of Abbreviation

| | |
|---|---|
| RHUG | ray-type hybrid underwater glider |
| CFD | computational fluid dynamics |
| HUG | hybrid underwater glider |
| AUV | autonomous underwater vehicle |
| DOF | degree of freedom |
| CG | center of gravity |
| PMM | planar motion mechanism |
| AHRS | attitude heading reference system |

## Nomenclature

| | |
|---|---|
| $u$ | surge velocity of the body-fixed frame |
| $v$ | sway velocity of the body-fixed frame |
| $w$ | heave velocity of the body-fixed frame |
| $p$ | roll angular velocity of the body-fixed frame |
| $q$ | pitch angular velocity of the body-fixed frame |
| $r$ | yaw angular velocity of the body-fixed frame |
| $\phi$ | Euler angle about $Ex$ axis in the earth-fixed frame |
| $\theta$ | Euler angle about $Ey$ axis in the earth-fixed frame |
| $\psi$ | Euler angle about $Ez$ axis in the earth-fixed frame |
| $x_g, y_g, z_g$ | coordinates of the center of gravity in the body-fixed frame |
| $x_b, y_b, z_b$ | coordinates of the center of buoyancy in the body-fixed frame |
| $X_{\dot{u}}$ | added mass coefficient in the surge motion caused by surge acceleration |
| $Z_{\dot{w}}$ | added mass coefficient in the surge motion caused by heave acceleration |
| $Z_{\dot{q}}$ | added mass coefficient in the surge motion caused by pitch angular acceleration |
| $M_{\dot{w}}$ | added mass coefficient in the pitch motion caused by heave acceleration |
| $M_{\dot{q}}$ | added mass coefficient in the pitch motion caused by pitch angular acceleration |
| $X_{uu}$ | damping coefficient in the surge motion caused by surge velocity |
| $X_{uw}$ | damping coefficient in the surge motion caused by surge and heave velocities |
| $X_{ww}$ | damping coefficient in the surge motion caused by heave velocity |
| $Z_{uu}$ | damping coefficient in the heave motion caused by surge velocity |
| $Z_{uw}$ | damping coefficient in the surge motion caused by surge and heave velocities |
| $Z_{ww}, Z_{www}$ | damping coefficients in the surge motion caused by heave velocity |
| $M_{uu}$ | damping coefficient in the pitch motion caused by surge velocity |
| $M_{uw}$ | damping coefficient in the pitch motion caused by surge and heave velocities |
| $M_{ww}, M_{www}$ | damping coefficients in the pitch motion caused by heave velocity |
| $N_{\dot{r}}$ | added mass coefficient in the yaw motion caused by yaw acceleration |
| $N_r, N_{|r|r}$ | linear and quadratic damping coefficients in the yaw motion |
| $I_{zz}$ | moment of inertia about $Oz_0$ axis |
| $\tau_u$ | control input induced by thrusters in the surge motion |
| $\tau_w$ | control input induced by buoyancy engines in the heave motion |
| $\tau_q$ | control input induced by moving mass in the pitch motion |
| $\tau_r$ | control input induced by thrusters in the yaw motion |

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
