# Peer review of "Robust Adaptive Heading Control for a Ray-Type Hybrid Underwater Glider with Propellers"

_jmse, doi:10.3390/jmse7100363_

Round 1

Reviewer 1 Report

This paper presents a controller for an underwater glider path planning (UGPP).

1. For UGPP, differential evolution to optimize the model parameters has been used lately, please comment on this and add some comparison:
Success history applied to expert system for underwater glider path planning using differential evolution. Expert Systems with Applications, 2019, vol. 119, pp. 155-170. DOI 10.1016/j.eswa.2018.10.048.
Constrained Differential Evolution Optimization for Underwater Glider Path Planning in Sub-mesoscale Eddy Sampling. Applied Soft Computing, 2016, vol. 42, pp. 93-118. DOI 10.1016/j.asoc.2016.01.038.
Differential Evolution and Underwater Glider Path Planning Applied to the Short-Term Opportunistic Sampling of Dynamic Mesoscale Ocean Structures. Applied Soft Computing, Vol. 24, November 2014, pp. 95-108. DOI 10.1016/j.asoc.2014.06.048.

2. Regarding evaluation of the system, please add some comparison method to show the reference performance effectiveness of the approach.

3. For experiment results, please add more than one simulation run and scenario, then provide aggregated overview estimation of robustness of the proposed approach in non-convex path planning scenarios.

Author Response

Dear Reviewer,

Thank you for your intensive comments.

Best regards,

Reviewer 2 Report

The manuscript studies the modelling of a ray-type hybrid underwater glider with propellers, and provides the theory and experimental tests of an adaptive robust controller for heading control. The manuscript starts with the description of the vehicle, its dynamic model and how to decouple its dynamics. It then proceeds to the estimation of the hydrodynamic parameters through a CFD analysis. Given that the exact value of the parameters is unknown, the authors provide the synthesis of a robust adaptive sliding mode controller for heading. That algorithm is tested with numerical simulations, water tank tests and tests at sea.

Although it has some shortcomings, given the completeness of the manuscript which provides a description of the mechanical setup, the complete dynamics model of the vehicle, the decoupling between vertical and horizontal motion, the CFD analysis, the controller synthesis considering disturbances and unknown parameters and experiments with the real vehicle at sea, I consider the manuscript to be a very valuable contribution to the field after some mistakes are corrected and the presentation is improved.

In the attached file suggestions on how to improve the manuscript are provided. 

The major concerns are the following:

One of the main concerns is the presentation, in particular regarding the english language and style. Minimal suggestions on how to improve these are given the attached file. I suggest the authors to proofread the manuscript with someone poficient in english.

Another concern is the fact that it is mentioned that the adaptive algorithm identifies the model parameters. This is not accurate since from the theoretical analysis only guarantees that the heading of the vehicle converges to close its assigned value, and does not provide any guarantees on the convergence of the estimated parameters to the real values. Therefore the estimated parameters can only be viewed as extra states of the controller which help in the heading control task. This fact should be clear when presenting the simulation and test results, since the fact that the estimated parameters do not converge to the real values should not be viewed as a problem of the adaptive controller.

Finally, there should be a description of the heading sensor used in the tests with the real vehicle.

Author Response

Dear Reviewer,

Thank you for your comments. I corrected my manuscript as your recommendation.

Best regards,

Reviewer 3 Report

It is a well-written paper; the argumentation and the presentation of both simulations and experiments are sound. Considering that this glider has significant applications, it is suggested the authors to explain further the acting and induced forces and their analysis. A paragraph and/or scheme could suffice.  

Author Response

Dear Reviewer,

Thank you for your encouraging comments.

Best regards,

Round 2

Reviewer 1 Report

The authors have not updated the manuscript according to the requested major revision comments. The answers to comments are merely descriptive as one can already read in the manuscript.
None of the 3 replies to revision comments are sufficient. To improve the paper with the requested revision, the paper should have been substantially improved.

If perhaps the comments have not been clear enough regarding the connection to glider vehicles, please note that the manuscript specifically writes that "the main objective of the RHUG hull design is gliding motion", hence the requested comments from first revision apply thorougly.

Provided the speed of progess from previous revision and weight of work required to address the provided comments in previous revision, I suggest this paper should be rejected and perhaps resubmitted at some rather later time when work is improved.

Author Response

Dear Reviewer,

Thank you for your comments. I answer your questions in the enclosure.

Best regards,
